DATA RELEASE

# Insecticide resistance dynamics in *Aedes* mosquitoes in Ghana

Isaac Kwame Sraku[1], Christopher Mfum Owusu-Asenso[1], Anisa Abdulai[1], Margaret Addo[1], Sebastian Kow Egyin Mensah[1], Faustina Adobea Owusu[1], Yaw Akuamoah-Boateng[1], Abdul Rahim Mohammed Sabtiu[1], Emmanuel Nana Boadu[1], Grace Arhin Danquah[1], Ruth Owusu Kwarteng[1], Cornelia Appiah-Kwarteng[2], Akua Obeng Forson[3], Simon Kwaku Attah[1] and Yaw Asare Afrane[1, *]

1 Centre for Vector-Borne Disease Research, Department of Medical Microbiology, University of Ghana Medical School, Korle-Bu, Accra GA-270, Ghana
2 School of Veterinary Medicine, University of Ghana, Legon, Accra, Ghana
3 Department of Medical Laboratory Science, School of Biomedical and Allied Health Sciences, University of Ghana, Korle-Bu, Accra, Ghana

## ABSTRACT

Arboviral diseases such as dengue, chikungunya, Zika, and yellow fever are of increasing endemicity and public health concern in Africa. Understanding the spatial distribution and dynamics of insecticide resistance in the *Aedes* vector could guide effective control interventions. We conducted larval surveys and WHO adult susceptibility bioassays on emerged adults from January 2019 to December 2023 in Ghana. Bioassays revealed widespread resistance in *Ae. aegypti* to pyrethroids, with 33.8–88.8% mortality for deltamethrin and 65–89% for permethrin. *Ae. aegypti* from Paga, Takoradi, and Accra was susceptible to pirimiphos-methyl. *Ae. vittatus* exhibited confirmed or possible resistance to pyrethroids. *Ae. albopictus* was found susceptible to all insecticides tested. Genotyping of mosquitoes (*n* = 887) identified high allelic frequencies of the F1534C kdr mutation in the pyrethroid-resistant *Ae. aegypti* populations. These findings highlight widespread pyrethroid resistance in the Ghanaian *Aedes* populations driven primarily by target-site insensitivity, and emphasize the urgent need for evidence-based vector-management strategies.

**Subjects** Ecology, Biodiversity, Taxonomy

**Submitted:** 22 September 2025

* Corresponding author. Email: yafrane@ug.edu.gh

Preprint submitted at https://africarxiv.ubuntunet.net/handle/1/10402

Included in the series: *Vectors of human disease* (https://doi.org/10.46471/GIGABYTE_SERIES_0002)

## BACKGROUND

*Aedes* mosquitoes are important vectors of several arboviruses, including dengue, chikungunya, Zika, and yellow fever, which pose a significant global public health problem in Africa. Dengue has emerged as one of the most rapidly expanding arboviral diseases worldwide. While historically concentrated in Southeast Asia and the Americas, dengue has increasingly been reported in Africa, raising significant concerns about its epidemiological shift. The continent is now experiencing frequent and intense outbreaks, with several countries reporting thousands of suspected and confirmed cases within short periods [1, 2].

Recent years have seen confirmed epidemics in countries such as Sudan, Kenya, Senegal, Burkina Faso, and Côte d'Ivoire [3, 4]. The true burden, however, remains underestimated due to limited diagnostic capacity, under-reporting, and frequent misclassification of dengue as malaria [5]. The current outbreaks highlight a convergence of risk factors, including rapid urbanization, inadequate vector control infrastructure, climate variability,

and the increasing adaptability of *Aedes aegypti* and *Aedes albopictus* as the primary vectors [6].

In the past decade, *Aedes* resistance surveillance in Africa, which has historically been underexplored in comparison to the Americas and Asia, has rapidly increased. Many control programmes are threatened by insecticide resistance in *Ae. aegypti* and *Ae. albopictus*. Studies have shown widespread pyrethroid resistance in *Ae. aegypti* and *Ae. albopictus* in several countries in studies from West, Central, and East Africa [7, 8]. Resistance phenotypes are mainly due to two broad mechanisms: target-site mutations in the voltage-gated sodium channel (VGSC) gene, referred to as knockdown-resistance (kdr), and metabolic detoxification via cytochrome P450 monooxygenases, esterases and glutathione S-transferases [9]. These mutations represent a critical genetic adaptation that enables these mosquitoes to survive insecticide exposure [8].

In West Africa, studies have shown high levels of phenotypic pyrethroid resistance in the *Aedes* population and the co-occurrence of kdr mutations (F1534C, V1016I, and V410L) [4, 7]. Although studies on arbovirus vectors in Africa are increasing [10], comprehensive data on phenotypic and genotypic resistance in *Aedes* species remain limited compared to malaria vectors.

Characterizing the phenotypic and genotypic insecticide resistance in *Aedes* mosquitoes is essential for designing sustainable vector management strategies. Integrating species distribution data with resistance profiles provides a more comprehensive understanding of local transmission dynamics and informs targeted interventions. Such studies are particularly important in endemic regions where insecticide use is high, and the burden of arboviral diseases is a significant public health concern.

## METHODS

### Study sites

We conducted a longitudinal study in 11 communities in three ecological zones in Ghana: the Sahel Savannah zone (Navrongo, Paga, Kpalsogu, Pagaza, Larabanga), the Forest zone (Wenchi, Konongo), and the Coastal Savannah zone (Ada, Accra, Tema, Takoradi) (Figure 1). The coastal savannah in the southern part of Ghana experiences a bimodal rainfall pattern, with a primary season from April to July, recording average rainfall of 1,000–2,000 mm. This season is followed by a short dry season in August, then a secondary rainy season from September to November, and finally the primary dry season from December to February [11].

The rainfall pattern in the forest zone of Ghana is characterized by considerable seasonal and interannual variability, high annual totals, and a longer rainy season than in other regions of the country. It has a tropical moist semi-deciduous climate with a characteristic double maxima rainfall pattern, with the primary rainy season peaking from May to July. The secondary rainy season from September to November ushers in the dry season from August, and the longer dry season from December to February is influenced by the Harmattan winds [12].

The Sahel Savannah zone of Ghana, which forms part of the broader West African Sahel zone, has a unimodal rainfall pattern characterized by significant spatial and temporal variability, quite different from that of the forest zone. A single rainy season from May to October peaking in August–September, is followed by a dry season from November to April, dominated by the Harmattan winds.



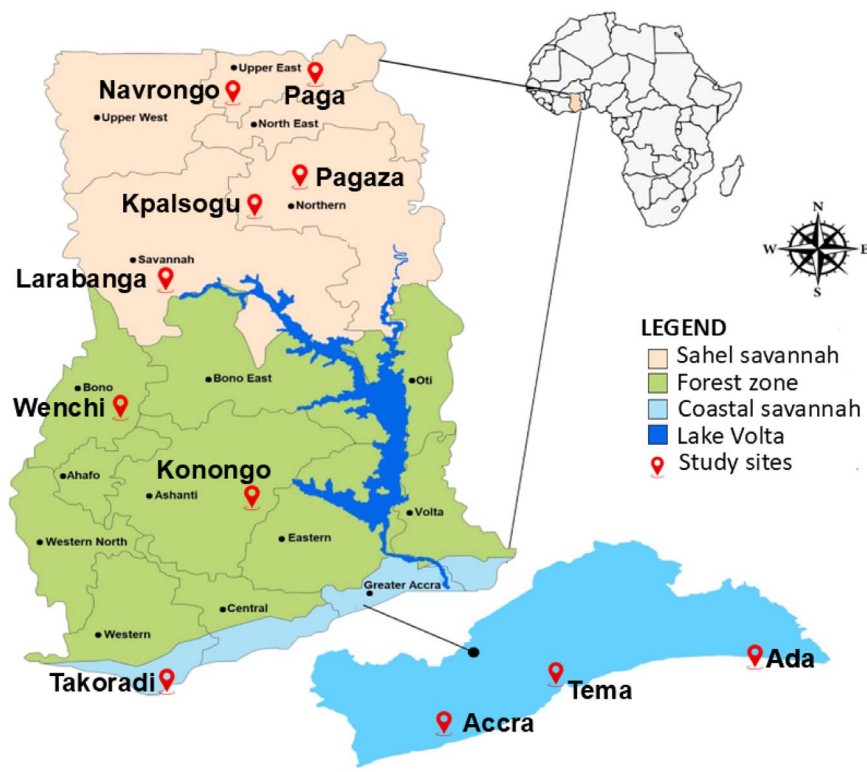

**Figure 1.** Map showing the various study sites.

## *Aedes* larval collection and adult identification

Larval collections were done between January 2019 and March 2025 during the rainy and dry seasons. Larval sampling was done using Plastic bowls, dippers and ladles for the collection of *Aedes* immature stages. Breeding habitats such as drains, discarded car tyres, household water storage containers, and deserted containers were inspected to collect immature *Aedes* stages. They were transported to the insectary and raised under controlled conditions of temperature (27 ± 2 °C) and relative humidity (75–80%) [13]. Emerged adult mosquitoes were provided with a 10% sugar solution as their primary energy source. Upon emergence, female *Aedes* mosquitoes were identified morphologically through direct visual inspection and confirmed to the species level using a stereomicroscope and taxonomic keys [14]. Three- to five-day-old, sugar-fed female mosquitoes were used for the World Health Organization (WHO) insecticide-susceptibility bioassay and genotypic analyses.

## Phenotypic resistance in *Aedes* mosquitoes using the WHO susceptibility tube bioassay

Three- to five-day-old sugar-fed adult female mosquitoes that emerged from field-collected larvae were subjected to the WHO susceptibility tube test to assess resistance using 150 mosquitoes per insecticide (25 mosquitoes per tube). Six tubes, consisting of four test replicates and two controls, were used for each insecticide, according to the WHO standard tube assay procedure [15]. WHO insecticide test papers of discriminating concentrations used were deltamethrin [(0.03%), (0.05%)], permethrin [(0.25%), (0.75%)], and



pirimiphos-methyl [(0.25%)] [15]. Knockdown was recorded at regular intervals during exposure for an hour, and mortality was scored after a 24-hour recovery period under insectary conditions with access to a 10% sugar solution. Mortality rates (MRs) were calculated and interpreted according to WHO criteria: MRs ≥ 98% indicated susceptibility, 90–97% suggested possible resistance requiring confirmation, and <90% confirmed resistance [15]. For data before 2024, the WHO insecticide-impregnated diagnostic concentrations for the determination of Anopheles resistance were used, as we did not have access to Aedes resistance determination concentrations. Subsequently, we obtained the WHO insecticide diagnostic concentrations for Aedes resistance determination and conducted the test to ensure accurate species-specific assessment.

Knockdown was recorded at regular intervals during exposure for an hour, and mortality was scored after a 24-hour recovery period under insectary conditions with access to a 10% sugar solution. MRs were calculated and interpreted according to the WHO criteria: MRs of ≥98% indicated susceptibility, 90–97% suggested possible resistance requiring confirmation, and <90% confirmed resistance [15].

## Genotypic resistance determination

Mutation in the VGSC that confers resistance to *Aedes* mosquitoes were determined using the protocols of Linss *et al.* [16] and Villanueva-Segura *et al.* [17]. Genotyping was done using conventional allele-specific PCR at V410, V1016, and F1534C positions of the VSGC. The allelic genotypic frequencies were calculated using the formula:

$$\text{Allelic frequency} = \frac{2RR + RS}{2n}$$

where *RR* is the number of homozygote mutants, *RS* is the number of heterozygotes, and *n* is the total number of mosquitoes analyzed.

## Data validation and quality control

All collected larvae and pupae were reared in the laboratory under standard procedures until adults emerged. The Geographical Position System of all collection points were taken.

Morphological identification of *Aedes* mosquitoes was performed by a trained entomologist using the identification keys by [14] to differentiate *Aedes* species.

The dataset was validated and published through the Global Biodiversity Information Facility (GBIF) Integrated Publishing Toolkit, which performs structural and content quality checks before release, with metadata fields available on the GBIF page to ensure transparency and facilitate reuse [18].

## RESULTS

## Phenotypic resistance

*Ae. aegypti* mosquitoes sampled from all the study sites except for Paga and Konongo showed resistance to deltamethrin (*MR* = 33.75–88.8%). *Ae. aegypti* from Paga and Konongo showed possible resistance to deltamethrin (*MR* = 92.5–93%). *Ae. aegypti* sampled from all sites except Kpalsogu, Pagaza and Paga were resistant to permethrin (*MR* = 65–89%), while they were partially resistant to permethrin in Kaplsogu, Pagaza and Paga (*MR* = 95–96.5%). *Ae. aegypti* mosquitoes from Kpalsogu and Konongo were resistant to pirimiphos-methyl (*MR* = 31.7–63.75%), whereas partial resistance was observed in Larabanga, Pagaza,

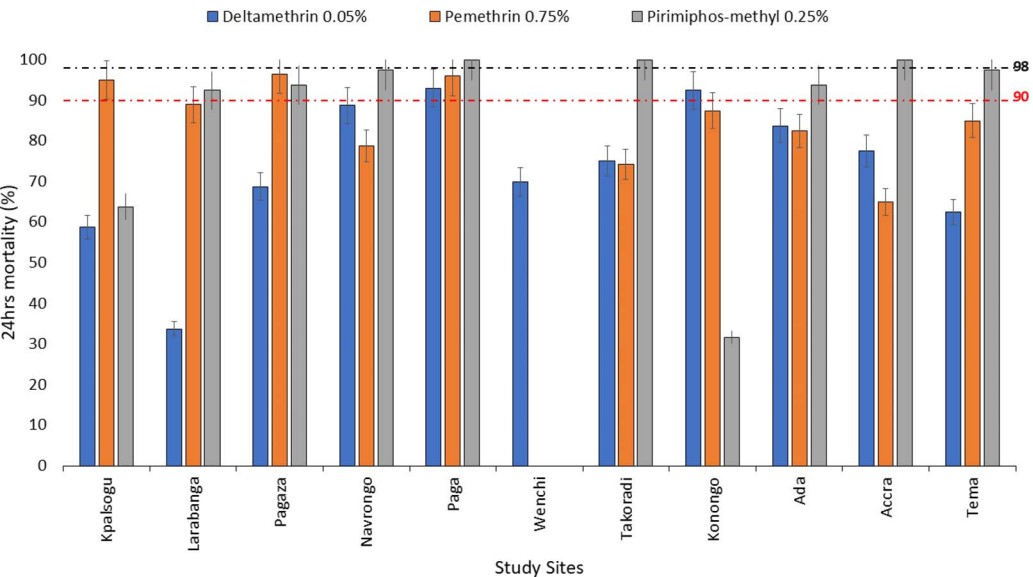

**Figure 2.** Twenty-four-hour post-exposure mortality of *Aedes aegypti* mosquitoes to different classes of insecticides across study sites.

Navrongo, Ada, and Tema (*MR* = 92.5–97.5%). However, *Ae. aegypti* sampled from Paga, Takoradi, and Accra were susceptible to pirimiphos-methyl (Figure 2).

*Aedes vittatus* mosquitoes sampled from Tema were resistant to pyrethroids (*MR* = 78.1–79.2%). *Ae. vittatus* sampled from the industrial area and Korle-Bu showed possible resistance to pyrethroids (*MR* = 91.25–97.5%). *Ae. albopictus* mosquitoes sampled from Takoradi were susceptible to all insecticides tested (Figure 3). *Ae. Albopictus* larvae from Takoradi were resistant to pyrethroids. *Aedes albopictus* larvae were not found in the other sites.

### Genotypic resistance of *Aedes* mosquitoes in the study sites in Ghana

A subset of 887 mosquitoes (689 *Ae. aegypti*, 114 *Ae. albopictus*, and 84 *Ae. vittatus*) obtained from the phenotypic assays were genotyped for the F1534C, V1016I, and V410L kdr mutations (Table 1). High allelic frequencies (*F*) of the 1534C kdr mutation in pyrethroid-resistant *Ae. aegypti* was detected in Konongo (*F* = 0.97) and Navrongo (*F* = 0.97), Ada (*F* = 0.96), Tema (*F* = 0.93), Wench (*F* = 0.90), Accra (*F* = 0.88), and Takoradi (*F* = 0.72). However, low frequencies of the 1534C kdr mutation were observed in *Ae. albopictus* (*F* = 0.346).

The 1534C kdr mutation frequency in *Ae. vittatus* was 0.32 in Korle-Bu and 0.05 in Accra. The V1016L mutation was also detected in all species at all the sites, with allelic frequencies ranging from 0.02 to 0.80. Finally, the V410 mutation frequency was 0.30 in *Ae. aegypti* mosquitoes sampled in Takoradi and 0.27 in *Ae. albopictus* mosquitoes sampled from Takoradi (Table 2).

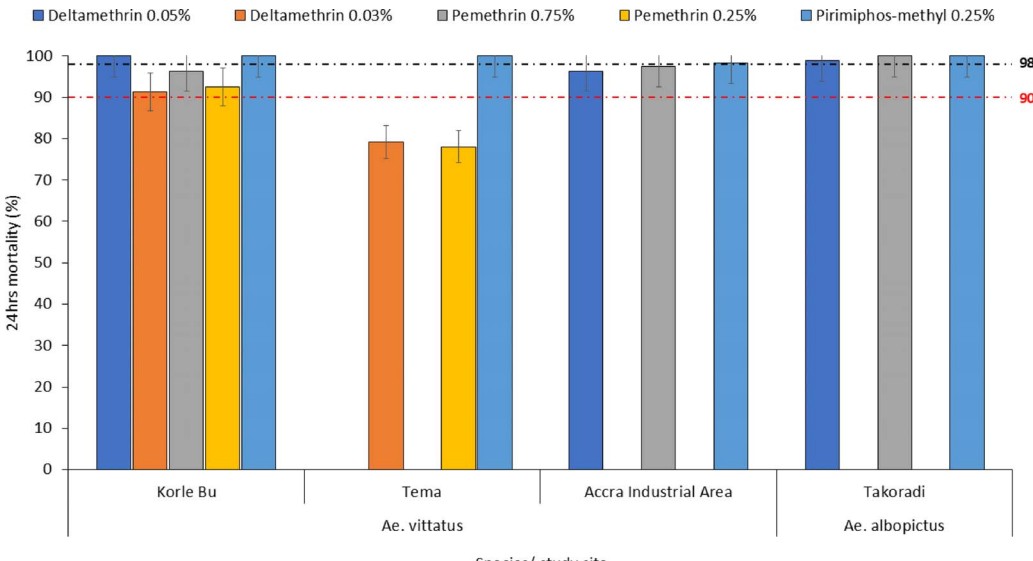

**Figure 3.** Twenty-four-hour post-exposure mortality of *Ae. vittatus* and *Ae. albopictus* mosquitoes to different classes of insecticides across study sites.

**Table 1.** Genotypes and frequencies of the F1534, V410, and V1016 mutations in the voltage-gated sodium channel gene of *Aedes* mosquitoes.

| Ecozones | Study site | Species | n | F1534C | | | | V1016l | | | | V410L | | | |
|---|---|---|---|---|---|---|---|---|---|---|---|---|---|---|---|
| | | | | RR | RS | SS | Allele Freq. | RR | RS | SS | Allele Freq. | RR | RS | SS | Allele Freq. |
| Sahel-savannah | Navrongo | | 69 | 67 | 0 | 2 | **0.97** | 52 | 7 | 10 | **0.80** | 6 | 3 | 60 | **0.11** |
| | Kpalsogu | | 44 | 12 | 7 | 25 | **0.35** | 1 | 42 | 1 | **0.50** | 3 | 1 | 40 | **0.08** |
| | Pagaza | | 92 | 14 | 14 | 64 | **0.23** | 3 | 50 | 39 | **0.30** | 2 | 16 | 74 | **0.11** |
| | Paga | | 0 | 0 | 0 | 0 | **0.00** | 0 | 0 | 0 | **0.00** | 0 | 0 | 0 | **0.00** |
| | Larabanga | | 54 | 13 | 12 | 29 | **0.35** | 15 | 39 | 0 | **0.64** | 0 | 3 | 51 | **0.03** |
| Forest zone | Wenchi | *Ae. aegypti* | 52 | 43 | 8 | 1 | **0.90** | 3 | 45 | 4 | **0.49** | 3 | 1 | 47 | **0.07** |
| | Konongo | | 17 | 16 | 1 | 0 | **0.97** | 3 | 0 | 14 | **0.18** | 0 | 6 | 11 | **0.18** |
| Coastal savannah | Ada | | 54 | 52 | 0 | 2 | **0.96** | 36 | 4 | 14 | **0.70** | 1 | 2 | 51 | **0.04** |
| | Tema | | 94 | 83 | 8 | 3 | **0.93** | 44 | 20 | 32 | **0.57** | 17 | 4 | 36 | **0.20** |
| | Accra | | 58 | 44 | 14 | 0 | **0.88** | 34 | 19 | 5 | **0.75** | 0 | 0 | 0 | **0.00** |
| | Takoradi | | 155 | 75 | 74 | 6 | **0.72** | 0 | 149 | 6 | **0.48** | 32 | 28 | 95 | **0.30** |
| | Other *Aedes* Species | | | | | | | | | | | | | | |
| | Takoradi | *Ae. albopictus* | 114 | 22 | 35 | 35 | **0.35** | 1 | 3 | 110 | **0.02** | 0 | 61 | 53 | **0.27** |
| Coastal savannah | Korle-Bu | *Ae. vittatus* | 44 | 14 | 0 | 16 | **0.32** | 0 | 0 | 44 | **0.00** | 9 | 0 | 20 | **0.20** |
| | Tema | *Ae. vittatus* | 0 | 0 | 0 | 0 | **0.00** | 0 | 0 | 0 | **0.00** | 0 | 0 | 0 | **0.00** |
| | Accra | *Ae. vittatus* | 40 | 2 | 0 | 0 | **0.05** | 0 | 0 | 34 | **0.00** | 10 | 0 | 30 | **0.25** |

## Species identification of *Aedes* mosquitoes in the various sites in Ghana

Out of the total 2,414 *Aedes* mosquitoes identified, the most abundant species were *Aedes aegypti* (2,204/2,414, 91%), followed by *Ae. albopictus* (114/2,414, 5%) and *Ae. vittatus* (96/2,414, 4%) (Table 2).



**Table 2.** Species identification of *Aedes* mosquitoes in the study sites in Ghana.

| Study Sites | *Aedes* species | | |
|---|---|---|---|
| | *Ae. aegypti* | *Ae. albopictus* | *Ae. vittatus* |
| Navrongo | 40 | 0 | 0 |
| Kpalsogu | 102 | 0 | 0 |
| Pagaza | 75 | 0 | 0 |
| Paga | 136 | 0 | 0 |
| Larabanga | 884 | 0 | 0 |
| Wenchi | 325 | 0 | 0 |
| Ada | 102 | 0 | 0 |
| Tema | 221 | 0 | 0 |
| Accra | 92 | 0 | 96 |
| Konongo | 72 | 0 | 0 |
| Takoradi | 155 | 114 | 0 |

## RE-USE POTENTIAL

Our study provides crucial insights into the evolving dynamics of insecticide resistance in the *Aedes* mosquito populations, establishing an essential baseline for longitudinal monitoring and vector control management in Ghana. The resistance to pyrethroid observed in *Ae. aegypti* and the highly invasive *Ae.* albopictus and *Ae. vittatus* aligns with previous reports from Ghana and other West African countries, where pyrethroid resistance has been consistently documented in both larval and adult populations [19]. Similar resistance trends have also been reported across sub-Saharan Africa and Asia, emphasizing the global scale of this challenge and the potential threat it poses to the efficacy of pyrethroid-based control tools [20, 21].

The detection of high frequencies of *kdr* mutations, *F1534C, V1016I,* and *V410L,* further supports earlier findings from Ghana, where these alleles have been linked to pyrethroid resistance in *Ae. aegypti* populations [19, 22]. However, our data indicate a slightly higher prevalence of the V410L mutation than previously reported by Amlalo *et al.* [22], suggesting local selection pressure by intense insecticide use. This observation may indicate global patterns, where co-occurrence of multiple *kdr* alleles has been associated with enhanced resistance phenotypes and operational control failures [23].

Low allele frequencies of *kdr* mutations were observed in *Aedes albopictus,* which has historically shown lower resistance levels. Recent studies from other African countries, such as Zambia and Cameroon, reported *Ae. albopictus* populations are adapting rapidly to anthropogenic environments and insecticidal exposure [24, 25]. The cumulative data from our study provide important data for *Aedes* vector control. National malaria control programs, neglected tropical diseases programmes, and municipal health authorities can leverage this resistance profile data to refine insecticide selection for adulticide campaigns and guide emergency and outbreak responses to arboviral diseases. This granular resistance data can significantly enhance the resilience of vector control programs against resistance-driven failures and support global efforts to curb the spread of arboviral diseases.

## DATA AVAILABILITY

The dataset supporting this study is available on the GBIF repository [26].



## EDITOR'S NOTE

This paper is part of a series of Data Release articles working with GBIF and supported by TDR, the Special Program for Research and Training in Tropical Diseases, hosted at the World Health Organization [27].

## ABBREVIATIONS

F, allelic frequencies; GBIF, Global Biodiversity Information Facility; kdr, knockdown-resistance; MR, mortality rate; VGSC, voltage-gated sodium channel; WHO, World Health Organization.

## DECLARATIONS

### Ethics approval and consent to participate

This study received scientific and ethical approval from the Ethical and Protocol Review Committee (EPRC) of the College of Health Sciences, University of Ghana, Korle-Bu Campus. Informed consent, verbal and written, was obtained from key opinion leaders at the various study sites prior to conducting mosquito sampling.

### Competing interests

The authors declared that there is no conflict of interest related to this research.

### Authors' contributions

IKS, CMO-A and YAA designed and supervised the study. IKS, CMO-A, AA, MOA, SKEM, MA, FAO YA-B, ARMS, GAD, CAK, AOF, and SKA were responsible for data collection, and contributed to the analysis of the data. IKS and CMO-A drafted the manuscript and data visualization. All the authors read and approved the final manuscript.

### Funding

This study was funded by grants from the National Institute of Health (R01 A1123074, R03 AI186018, and D43 TW011513). The funders had no role in the study design, data collection, analyses, interpretation, or manuscript preparation.

### Acknowledgements

We express sincere gratitude to the community members and field assistants for their support and cooperation during sample collection and for granting permission to conduct the study in their localities.

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
