## [Editor Report]

Editor’s AssessmentIn Sub Saharan Africa Arboviral diseases such as dengue, chikungunya, Zika, and yellow fever are of increasing endemicity and public health, with dengue representing a rapidly growing threat. Understanding the spatial distribution and insecticide resistance dynamics of Aedes vectors is critical for guiding effective control interventions but there is limited data on their spatiotemporal distribution, which hinders control strategies. This paper is one of a series of Data Release papers in GigaByte supported by TDR and the WHO describing datasets hosted in GBIF to tackle these data gaps in vectors of human disease data. This paper presents a longitudinal study from January 2019 to December 2023 sampling 11 communities in three ecological zones in Ghana comprising of the Sahel Savannah, Forest zone, and Coastal Savannah. Bioassays and genotyping of resistant mosquitoes revealed widespread resistance, provides key insights into the current dynamics and establishing a vital baseline for longitudinal surveillance. Peer review and data auditing found the data to be well validated. The information contained can serve as a resource for studies focused on assessing transmission risks, vector control strategies, disease surveillance and a broader comprehension of mosquito ecology and insecticide resistance in the various ecological zones of Ghana.Editor’s AssessmentIn Sub Saharan Africa Arboviral diseases such as dengue, chikungunya, Zika, and yellow fever are of increasing endemicity and public health, with dengue representing a rapidly growing threat. Understanding the spatial distribution and insecticide resistance dynamics of Aedes vectors is critical for guiding effective control interventions but there is limited data on their spatiotemporal distribution, which hinders control strategies. This paper is one of a series of Data Release papers in GigaByte supported by TDR and the WHO describing datasets hosted in GBIF to tackle these data gaps in vectors of human disease data. This paper presents a longitudinal study from January 2019 to December 2023 sampling 11 communities in three ecological zones in Ghana comprising of the Sahel Savannah, Forest zone, and Coastal Savannah. Bioassays and genotyping of resistant mosquitoes revealed widespread resistance, provides key insights into the current dynamics and establishing a vital baseline for longitudinal surveillance. Peer review and data auditing found the data to be well validated. The information contained can serve as a resource for studies focused on assessing transmission risks, vector control strategies, disease surveillance and a broader comprehension of mosquito ecology and insecticide resistance in the various ecological zones of Ghana.

---

## [Reviewer Report]

Indicate in the comments box below whether you are happy with the changes made or if the manuscript is unacceptable.Comments on revised manuscriptLine 113: Please revert to the initial title, Phenotypic Resistance of Aedes Mosquitoes Using the WHO Susceptibility Tube Assay. If you intend to include “intensity” in the title, you must describe in the Methods section how you applied the 2×, 5×, and/or 10× diagnostic doses and present the corresponding results. These details are currently not provided in the manuscript. Apart from this, the remaining issues are minor typographical errors and the incorrect formatting of scientific names, which I have annotated in the attached document (via email—couldn't attach it here). Overall, I am satisfied with the revisions and recommend acceptance of the manuscript.

---

## [Reviewer Report]

Reviewer name and names of any other individual's who aided in reviewer Yannan FanDo you understand and agree to our policy of having open and named reviews, and having your review included with the published papers. (If no, please inform the editor that you cannot review this manuscript.)YesIs the language of sufficient quality?YesPlease add additional comments on language quality to clarify if needed
Are all data available and do they match the descriptions in the paper? YesAdditional CommentsAre the data and metadata consistent with relevant minimum information or reporting standards? See GigaDB checklists for examples <a href="http://gigadb.org/site/guide" target="_blank">http://gigadb.org/site/guide</a>YesAdditional CommentsIs the data acquisition clear, complete and methodologically sound?YesAdditional CommentsIs there sufficient detail in the methods and data-processing steps to allow reproduction?YesAdditional CommentsIs there sufficient data validation and statistical analyses of data quality? YesAdditional CommentsIs the validation suitable for this type of data?YesAdditional CommentsIs there sufficient information for others to reuse this dataset or integrate it with other data?YesAdditional CommentsAny Additional Overall Comments to the AuthorPlease update the GBIF reference as: Afrane Y A, Owusu-Asenso C M, Akuamoah-Boateng Y, Abdulai A, Mohammed Sabtiu A R (2025). Phenotypic and Genotypic Insecticide Resistance Profiles of Aedes Mosquitoes across Ghana. Version 1.1. University of Ghana Medical School. Occurrence dataset https://doi.org/10.15468/ef5mry accessed via GBIF.org on 2025-10-11.RecommendationAccept

---

## [Reviewer Report]

Upload additional filesDRR-202509-05-R02/stage_files/DRR-202509-05/Review MS/gx-DR-1758550957_Edited.pdfReviewer name and names of any other individual's who aided in reviewer Udoka NwangwuDo you understand and agree to our policy of having open and named reviews, and having your review included with the published papers. (If no, please inform the editor that you cannot review this manuscript.)YesIs the language of sufficient quality?YesPlease add additional comments on language quality to clarify if needed
However, while the manuscript is generally understandable it may require language editing for clarity and conciseness. Redundant phrasing should be minimized and standard entomological expressions (e.g., “immature stages”) used consistentlyAre all data available and do they match the descriptions in the paper? YesAdditional CommentsAre the data and metadata consistent with relevant minimum information or reporting standards? See GigaDB checklists for examples <a href="http://gigadb.org/site/guide" target="_blank">http://gigadb.org/site/guide</a>YesAdditional CommentsIs the data acquisition clear, complete and methodologically sound?NoAdditional CommentsMethodological details are insufficient. The authors should specify the number of mosquitoes per assay, control group composition, validation according to WHO protocol, determination of allelic frequencies and justify combining diagnostic doses for Anopheles and Aedes species in some areas and not others.Is there sufficient detail in the methods and data-processing steps to allow reproduction?NoAdditional CommentsReproducibility is limited by missing procedural details. Sampling, collection tools (dippers, pipettes, ladles), determination of allelic frequencies and specific WHO procedures should be clearly described for transparency and repeatability.Is there sufficient data validation and statistical analyses of data quality? YesAdditional CommentsHowever, the authors should include WHO standards as part of the data validationIs the validation suitable for this type of data?YesAdditional CommentsThe missing part of the validation steps must be explicitly described.Is there sufficient information for others to reuse this dataset or integrate it with other data?YesAdditional CommentsAny Additional Overall Comments to the AuthorThis study addresses a relevant and timely topic but requires major revision to meet GigaByte’s methodological, reporting and reproducibility standards. The discussion should be expanded with comparative literature.RecommendationMajor Revision